# Intergenerational Transmission of Human Parenting Styles to Human–Dog Relationships

**DOI:** 10.3390/ani14071038

**Published:** 2024-03-28

**Authors:** Chih Hsin Kuo, Sharon Kessler

**Affiliations:** 1College of Medical, Veterinary and Life Sciences, University of Glasgow, Glasgow G12 8QQ, UK; 2Division of Psychology, Faculty of Natural Science, University of Stirling, Stirling FK9 4LA, UK; sharon.kessler@stir.ac.uk

**Keywords:** intergenerational transmission, dog-directed parenting style, human–dog interaction, training method

## Abstract

**Simple Summary:**

This study explores how parenting styles toward dogs are influenced by experiences in one’s own upbringing. Through a mixed methods approach involving surveys from 391 dog owners and interviews with 10 participants, this study found that a permissive parenting style tends to be passed down across generations, with individuals who experienced it growing up being more likely to use it with their dogs. Orientation towards dogs played a key role, with a protectionistic orientation reducing the likelihood of authoritarian caregiving. Humanistic and protectionistic attitudes increased the chances of compensatory permissive behaviors. Insights from interviews emphasized the impact of childhood experiences on adopting specific parenting behaviors toward dogs. By drawing on our understanding of child–parent relationships, addressing the underlying elements of human–dog dynamics may lead to positive outcomes for both dogs and their caregivers.

**Abstract:**

Parenting style and intergenerational transmission have been extensively studied in parent–child relationships. As dogs are increasingly recognized as integral members of the family system, there is a growing interest in understanding how parenting behaviors directed towards dogs can also influence a dog’s behaviors. However, the reasons why people adopt certain parenting behaviors towards dogs remain relatively unknown. This study delved into the intergenerational transmission of parenting styles from one’s upbringing to caregiving for dogs. Using a mixed methods approach with 391 dog caregivers and 10 interviews, this study employed multivariate linear regression and thematic analysis. Permissive parenting exhibited an intergenerational effect, with those experiencing it being more likely to replicate the style with their dogs. Orientation towards dogs emerged as a crucial mediator, with protectionistic attitudes reducing the likelihood of replicating authoritarian parenting. Humanistic and protectionistic orientation increased the likelihood of compensatory permissive behaviors. Insights from interviews underscored the impact of perceived childhood experiences on adopting specific parenting behaviors. Ultimately, this study provides valuable insights that can contribute to the promotion of appropriate caregiving behaviors toward dogs. By drawing on our understanding of child–parent relationships, addressing the underlying elements of human–dog dynamics may lead to positive outcomes both for dogs and their caregivers.

## 1. Introduction

Over the past decade, there has been a notable surge in the number of pet dogs worldwide. Presently, estimates suggest that there are approximately 700 million dogs worldwide, with roughly 75% of them classified as free-roaming dogs. These free-roaming dogs vary in terms of guardianship status, with some having unrestricted movement under human care, while others are looked after by the community [1]. Among this vast dog population, it is estimated that around 470 million dogs are kept as household pets worldwide [2]. In the UK, roughly 13 million dogs live with their human caregivers [3], and in 2018, more than 46.1% of the U.S. population had dogs as pets [4]. Data on the pet dog population in Asia are relatively limited; however, approximately 30.9% of Korean households and 50% of Filipino families own one or more dogs [5,6]. As the number of multi-species households has increased, scholars have suggested that pet keeping practices in families have shifted to a post-humanist form [7]. According to this perspective, the family concept has broadened, from human-centered relationships to integrating nonhumans into kinship practices, forming multi-species households.

Building upon this model, Power [8] further proposed the “more than human” model, which emphasizes that dogs not only become incorporated into families but also actively participate in and influence everyday practices. Supporting this idea, studies focusing on human physiological responses report spillover effects of affiliative behaviors that originally evolved same-species caregiving contexts. For instance, when mothers interact with both their children and family dogs, similar patterns of increasing oxytocin levels and brain activation are elicited [9,10]. Even individuals who do not explicitly identify themselves as parents often include their pets in family photos and engage in behaviors like talking to their pets and speaking on their behalf in certain circumstances [11,12,13]. These actions reflect the deep emotional connection and sense of companionship that people feel towards their pets, blurring the boundaries between human and non-human animal relationships. Such behaviors demonstrate the ways in which pets are integrated into the fabric of family life and are considered important members of the household, with whom meaningful interactions and communications take place [14].

Despite the possibility that intimacy and interconnection between humans and dogs appears to be growing closer, it is critical to note that the majority of human–dog caregiving practices involve dogs being kept in a captive environment and being highly dependent on humans to meet their needs [15,16]. This dependency includes access to key resources such as food and water, timing to eliminate, and opportunities to explore. Stafford [17] noted that among all the factors that could potentially influence the welfare of a dog, the caregiver’s behaviors are likely to be among the most impactful. Furthermore, within the framework of One Health, One Welfare, the welfare of companion animals is intricately linked to both human wellbeing and the environment. Inappropriate caregiving practices can compromise the welfare of companion animals, potentially resulting in the escalation of undesirable behaviors and posing risks to their health. Such outcomes can have adverse effects on human and non-human animal interactions and may even lead to mental distress among caregivers. Additionally, these undesirable behaviors extend their impact to others in society, including neighbors, other individuals, and animals encountered in various social settings [18]. The increasing recognition of the significance of human and non-human animal interactions on the wellbeing of both humans and non-human animals underscores the importance of understanding the factors that shape caregivers’ dog-directed parenting behaviors. Given the context of unequal power dynamics in which both children and dogs heavily rely on adult family members, creating a similar dynamic that blurs the human and non-human animal boundary [19], we study the parallels between the parent–child relationship and the human–dog relationship. In this study, we investigate the potential intergenerational transmission effects of parenting styles, being transferred from human contexts (parent–child) to multi-species contexts (human–dog). The findings from this research may provide valuable insights into enhancing canine welfare and fostering stronger bonds between humans and their canine companions.


**Parenting Style**


Parenting styles refer to the way in which parents interact with their children and can significantly impact children’s development and mental wellbeing [20]. Likewise, as people often refer to their dogs as “fur babies” and view them as “surrogate children” [21], parenting behaviors and styles can also be identified in the human–dog relationship. Parenting behaviors can be classified into four main styles, based on variations in levels of demandingness and responsiveness. Demandingness refers to control and monitoring in parenting behaviors. Responsiveness is about emotional warmth and the ability to recognize other’s needs and provide support [22]. Depending on the variation in the dimensions of demandingness and responsiveness, four main parenting styles have been identified: authoritative, authoritarian, permissive, and uninvolved [22,23]. The authoritative parenting style is characterized by high demandingness and responsiveness, where behaviors are monitored with expectations, and goals and values are communicated between parents and their children. In contrast, the authoritarian parenting style is characterized by a coercive use of power, where strict adherence to rules is expected without clear explanations provided. This style tends to emphasize parental authority and hierarchical relationships, with limited room for negotiation or autonomy on the part of the child and low responsiveness to the child’s needs. The emphasis is on obedience rather than understanding, fostering an environment where conformity to rules is the primary focus [24]. Permissive parenting style involves high responsiveness but low demandingness, with parents being affectionate and recognizing their children’s needs but lacking the ability to provide direction and structure [25]. Finally, the uninvolved parenting style reflects low levels of both demandingness and responsiveness, representing the lowest effort parenting style [26,27].

Studies have suggested that the authoritative parenting style is the optimal parenting style [28], whereas an authoritarian parenting style has been associated with increased behavioral issues and emotional maladjustment in children. Increased behavioral problems, such as substance abuse and school misconduct in children, have also been suggested to be related to permissive parenting style [29]. Similarly, different dog-directed parenting styles have been found to influence dogs’ welfare and behaviors. Recent evidence suggests that distinct human–dog interactions and the dog’s attachment type, sociability, and problem-solving behaviors are associated with different dog-directed parenting styles [30,31]. When caregivers’ parenting behaviors are perceived as higher in warmth, dogs exhibit increased proximity-seeking behaviors in threatening situations [32]. Furthermore, permissive dog parenting has been associated with dogs being overweight [33]. Caregivers also exhibited a closer attachment to dogs kept indoors compared to those kept outdoors in yards, with indoor dogs receiving more attention and enrichment from their caregivers [34]. Overall, there is a growing body of research on human–dog interactions and their influence on human and dog behavior and physical and mental health; however, a comprehensive understanding regarding how people develop their dog-directed parenting style is still lacking. This gap highlights the need for further research and investigation to delve deeper into the nuances of dog-directed parenting styles and their implications for the wellbeing of both humans and dogs.


**Intergenerational transmission**


Previous research has suggested that parenting behaviors not only influence children’s development but also affect how their children will interact with the next generation [35,36]. Intergenerational transmission of parenting is recognized as a process whereby shared beliefs and behaviors are passed down from one generation to the next [37]. Intergenerational transmission of parenting has been identified in various studies across different cultures and sociodemographic backgrounds [38,39]. Although most studies focus on negative parenting behaviors and compromised development in children, such as maltreatment, chronic stress, and trauma, many studies have identified that transmission also occurs with warm and supportive parenting [40,41,42]. Furthermore, intergenerational transmission has been recognized among various sociodemographic backgrounds; it is suggested that the association is a bidirectional influence. Different cultures are characterized by their own unique beliefs and parenting practices, which can impact people’s perception of parenting responsiveness and demandingness [38]. Gender and marital status can also influence intergenerational transmission [39,43,44]. In the context of non-human animals, some scholars have argued that human and non-human animal relationships can mirror the dynamics observed in parent–child relationships [45,46,47]. Furthermore, it has been suggested that parental personality traits and parenting styles influence their interactions with non-human animals [45,48]. 


**Discontinuity of Parenting Behaviors**


Alongside intergenerational transmission, discontinuity in parenting styles has also been identified. Several factors have been recognized as contributing to discontinuity of parenting behaviors, i.e., gender differences, social–cultural shifts, increasing levels of education, marital status, and parents’ parenting behaviors [49,50,51]. An additional form of discontinuity in parenting style is compensation, which has been identified in child psychotherapy [52,53]. Compensation occurs when parents project the unmet needs from their own childhoods onto their children and try to prevent their children from having the same experiences they had [53,54]. This projection of unfulfilled needs might lead to overcompensating and the inability to recognize their child’s real needs and temperament, resulting in overcontrolling or overprotective parenting behaviors [54,55]. Such overcompensating parental behaviors also play a pivotal role in the significant association between early maladaptive schemas of parents and their adult children [56], which are self-defeating emotional and cognitive patterns developed in childhood [57]. Pets, being included in the intersecting relationships that constitute a family system [58], akin to children, can fulfil the role of a mental stabilizer within the family system, utilizing processes such as displacement, projection, and identification [59]. Additionally, the human and non-human animal bond has been hypothesized to serve as compensation for deficient human attachment, with a stronger attachment to pets potentially compensating for unmet needs for closeness, security, and emotional and social support within the parent–child relationship [60]. According to Ribera et al. [61], a child’s strong attachment to a dog might serve as a compensatory mechanism for an insecure attachment to the mother, leading to a protective role associated with better psychological adjustment during middle childhood.


**Orientation toward Non-Human Animals and Parenting styles**


A person’s orientation toward non-human animals is also recognized as an influential variable when considering human and non-human animal relationships. Increased positive attitudes toward pets have been associated with a more intimate relationship and a greater responsibility to provide daily care [62]. Moreover, attitudes toward dogs are significantly related to attachment, investment, abuse, and anthropomorphism toward dogs [63]. Blouin [64] classified the orientation toward non-human animals into three categories: dominionistic, humanistic, and protectionistc. Dominionistic attitudes emphasize the hierarchy in human–dog relationships, viewing dogs as less important than humans, where dogs are “objects” rather than “subjects”. Humanistic caregivers value their dog anthropomorphically, considering themselves as a parent or a friend of the dog and treating the dog as if were a human. Protectionistic caregivers view their dogs as companions but understand that dogs have species-specific needs. These orientations reflect differing dimensions of perceptions about the status of dogs. Building on Blouin [64], van Herwijnen et al. [65] further discovered that the authoritative-intrinsic value-oriented parenting style, which emphasizes being responsive to the real needs of the species, is strongly associated with a combination of humanistic and protectionistic orientation towards non-human animals. On the other hand, the authoritarian correction-oriented parenting style, characterized by high demandingness and a focus on using corrective techniques for dog behavior, is related to the dominionistic orientation towards non-human animals. These findings suggest that people’s orientation towards non-human animals might influence their dog-directed parenting style.

Despite abundant evidence describing the influence of parenting style on children’s development and its generational effect, little is known about the dog-directed parenting style and its potential influence on dog behavioral development and welfare. Moreover, the linkage of critical factors on dog development, such as orientation toward non-human animals and the association with parenting styles, remains relatively unknown. This study employed a mixed-methods approach to investigate the influence of individuals’ parenting behaviors towards their dogs, considering the parenting styles they received from their parents and the mechanisms involved. The first part of this study employed quantitative analysis. We aimed to investigate whether intergenerational transmission occurred between participants’ received parenting behaviors in childhood (referred to as received parenting style) and their own dog-directed parenting. Additionally, demographic factors were explored, to see if they played a role between received parenting style and dog-directed parenting style, based on previous research suggesting their potential impact [38,43,49]. Moreover, this study delved into exploring potential mediating effects between intergenerational transmission and orientation towards dogs. The core concepts of this study are shown in Figure 1.

Rodriguez et al. [66] highlighted that individual differences play a crucial role in shaping how people perceive, interact with, and form bonds with their dogs. In the second part of this study, we employed qualitative analysis in order to capture the richness and depth of participants’ experiences and perspective. This method allowed us to delve into the nuances of how the participants themselves interpret their human–dog relationships and gain a deeper understanding of the intergenerational transmission of dog-directed parenting styles. Via qualitative analysis, we explored the various factors that contribute to the formation of these relationships, such as participants’ upbringing, cultural background, and personal experiences.

## 2. Materials and Methods

This study utilized a mixed method approach. The quantitative components involved a survey administered online, focusing on whether intergenerational transmission occurs in dog parenting and whether orientation towards dogs plays a mediating role. In addition, the qualitative part entailed conducting semi-structured interviews. This provided an opportunity to delve into the interactions and relationships between the participants and their parents during childhood and to understand whether there is a tendency for caregivers to consciously choose to replicate or compensate with the parenting behaviors they direct towards their dogs. 


**Participant Recruitment**


Ethical approval for this study was granted from the University of Stirling General University Ethic Panel [Protocol #: 14044, date: 5 April 2023]. The study was open to participants worldwide, and recruitment took place online via various social media platforms, including Facebook groups and Instagram, from 5 April 2023 to 18 April 2023. Prior to participation, participants were presented with a consent form at the beginning of the survey. Eligibility criteria included being age 18 or above, having no diagnosis of mental illness and to be currently caring for at least one dog, regardless of the duration of that caregiving. Those interested in participating in the interview provided their email address and were sent a separate interview consent form. All interviews were conducted with the participants’ permission and audio was recorded.

Guest et al. (2006) [67] suggested hat six to twelve interviews are sufficient for developing meaningful themes and interpretations, we aimed to ensure a diverse sample size that would effectively address the study objectives. To achieve this, we randomly selected 10 participants from the pool of participants who had responded that they were willing to be interviewed and had signed the consent form for the interview.


**Instrumentation**


The survey was conducted using Testable (www.testable.org, accessed on 09 March 2023) with 104 questions (Appendix A), including 6 demographic questions, a 32-item Parenting Styles and Dimensions Questionnaire (32-PSDQ) [27], an adapted 31 item dog-directed parenting style form 33 questions [68] for understanding the orientation toward non-human animals [65] (van Herwijnen et al., 2020), and the final 2 questions asking about the participant’s willingness to participate in the interview. Testable, initiated in 2015, is a web platform providing a range of tools designed for psychology research purposes [69]. The survey was conducted under the University of Stirling’s license.

*Demographic questions:* Six questions asking the participant’s age, gender, education level, nationality, and whether they have experience caring for children and dogs.

*Thirty-two-item PSDQ form:* The 32-item short version of the Parenting Styles and Dimensions Questionnaire (PSDQ) was employed to assess different parenting styles. This shortened version, developed by Robinson, Mandleco, Olsen, and Hart [27], consists of 32 items and is based on Baumrind [70] authoritative, authoritarian, and permissive parenting styles. Participants rated each item on a 5-point Likert-type scale ranging from 1 (never) to 5 (always). PSDQ has been widely used in various countries and with large sample sizes [27,71,72]. It has demonstrated good validity in different contexts, including self-reports, spousal reports, and retrospective reports given to children to reflect on their parents [73]. PSDQ subscales have been reported with average Cronbach’s α values of 0.86 for authoritative, 0.82 for authoritarian, and 0.64 for permissive parenting styles [27].

*The Dog-Friendly Version PSDQ Form:* This questionnaire was adapted from van Herwijnen, van der Borg, Naguib, and Beerda’s [68] “The existence of parenting styles in the owner-dog relationship”. It contains 13 items reflecting the authoritative parenting style, 12 items related to the authoritarian, and 6 items reflecting the permissive parenting style in the context of dog caregiving practices. Consistent with the original PSDQ form, it was measured using a 5-point Liker-type, rating the likelihood of scenarios occurring as 1 (never) to 5 (always).

In our effort to validate the 32-item PSDQ form for assessing dog-directed parenting styles, we carefully selected items that corresponded to questions in the original 62-item construct developed by van Herwijnen, van der Borg, Naguib, and Beerda [68]. For example, the original statement concerning authoritative parenting style, “My parents provided me reasons why rules should be obeyed”, was matched with “I think about why rules should be obeyed by my dog” in the dog-directed PSDQ form. We excluded two items that were deemed incompatible with dog-directed parenting scenarios, including one authoritative-related item (“My parents explained the reasons for rules”) and one authoritarian-related item (“My parents openly criticized me when my behavior does not meet their expectations”). Additionally, to ensure a balanced representation of parenting styles in the dog-directed context, we introduced one item “I am struggling to try to change how my dog behave”, which is related to permissive parenting style.

*Orientation towards Non-Human Animals:* The remaining 33 questions were derived from van Herwijnen, van der Borg, Naguib, and Beerda’s [65] “Dog-Directed Parenting Styles Mirror Dog Owners’ Orientations Toward Animals”, which focused on participants’ orientations toward dogs across various aspects. These questions were based on a qualitative study by Blouin [64] and aimed to understand participants’ attitudes and perspectives toward non-human animals. Given the multifaceted nature of factors, which include financial constraints and health considerations that may supersede one’s orientation toward animals, items associated with relinquishment were excluded.

*Online Interview:* The interview portion involved conducting semi-structured interviews with participants via Microsoft Teams. The interviews lasted approximately 30 min to 1 h and consisted of 20 open-ended questions (Appendix B). Participants in the study were queried about their views on the human–dog relationship and their experiences of parenting behaviors during their childhood. The questions were aimed at exploring two dimensions of parenting style: how their parents responded to distress (responsiveness) and whether they had rules and routines to follow with associated consequences for violations (demandingness). Additionally, the study inquired about the opportunities for exploration during their childhood, which is considered an important aspect of the parent–child relationship [74]. The same set of questions was then presented in the context of dog caregiving practices, aiming to understand participants’ dog-directed parenting behaviors under similar circumstances. Participants were also asked about how they perceived any intergenerational transmission of their partner’s parenting style and whether their partner’s interactions with their dog impacted their own parenting behaviors. Lastly, participants were asked to identify any factors that may have influenced their interactions, such as mental health conditions, concerns about the dog’s behavior and health, and other unlisted factors. All interviews were audio recorded for transcription. 


**Data Analysis**


The quantitative data collected in this study were analyzed using SPSS (version 24.0). Descriptive statistics were employed to analyze the demographic information provided by participants. The average scores for PSDQ form, the Dog-Friendly Version PSDQ Form, and orientation toward non-human animals were calculated. Although Baumrind [70] originally advocated the categorization of parenting styles into three types, the recent literature has suggested examining parenting styles as continuous variables to address certain limitations, as the mean scores of the three parenting styles tend to be close [75,76]. No averages tied; therefore, the participant was classified under the parenting style or orientation toward dogs on which they had the highest score.

To investigate intergenerational transmission effects on parenting styles, multivariate linear regression analyses were conducted. Received parenting styles, along with demographic factors, were treated as fixed factors, while dog-directed parenting style was set as the dependent variable. Additionally, stratified analyses were conducted based on demographic factors. Owing to the constrained sample size, all Asian countries were amalgamated into a single group for analytical purposes. The regrouping of demographic factors was performed to ensure a balanced sample size across categories. In this analysis, the fixed factor was the received parenting styles, and the dependent factor was the dog-directed parenting style. The goal of the stratified analyses was to identify any significant associations or differences between received parenting styles and dog-directed parenting styles across various demographic subgroups. Some categories, such as age, did not have high enough sample sizes to be tested separately, so they were either combined into larger groups or excluded. Bonferroni corrections were implemented. The predetermined significance level was set at *p* < 0.05, with a Bonferroni-corrected level of *p* < 0.0167, accounting for three groups based on nationality and dog caring experience. In order to ensure the validity of our data, we performed comprehensive checks on the assumptions of multivariate linear regression. Normal predicted-probability plots were used to assess the normality of the residuals. Scatter plots were employed to examine the linearity of the relationship between the independent and dependent variables. Additionally, we checked the Durbin–Watson statistic to assess the presence of autocorrelation in the residuals, and the variance inflation factor (VIF) to detect multicollinearity among the independent variables. These analyses confirmed that our data met all the necessary assumptions for conducting reliable multivariate linear regression (Appendix C).

Baron and Kenny’s [77] method was utilized to explore whether orientation towards dogs mediates the relationship between the parenting style they received from their parents and their dog-directed parenting style. Mediation analysis was conducted using the Process v4.3 for SPSS. Likewise, received parenting styles were defined as the independent variable (X) and dog-directed parenting styles as the dependent variable (Y). In accordance with the sequence illustrated in Figure 2, we estimated three paths, namely path a, path b, and path c. These coefficients of these paths represent the strength and direction of the relationships between the variables in the mediation analysis. A positive coefficient means that an increase (decrease) in the independent variable is associated with an increase (decrease) in the mediator and, in turn, an increase (decrease) in the dependent variable, indicating a mediating effect that enhances or amplifies the relationship between the independent and dependent variables. While a negative coefficient suggests a suppressor effect, an increase in the independent variable is associated with a decrease in the mediator and, in turn, a decrease in the dependent variable. A 5000-bootstrap procedure was applied to generate the bias-corrected 95% confidence intervals (CI) to determine the indirect effect. The indirect effect was significant when the upper boundary and the lower boundary of the 95% bootstrapped confidence intervals do not contain zero [78]. Note that the indirect effects are not mapped in Figure 2.

Qualitative data were analyzed using thematic analysis following the Braun six-step method proposed by Braun and Clarke [79]. This approach, as outlined by Braun and Clarke, involved becoming familiar with the data, generating initial codes, grouping codes into potential themes, reviewing and refining themes, naming and renaming themes, and finally selecting the quotes that best represented each theme. Thematic analysis is a widely used inductive approach that involves disassembling and reassembling data and evaluating and interpreting it. All themes derived from the analysis were thoroughly examined and presented in the findings.

In accordance with the Standards for Reporting Qualitative Research [80] and with Braun et al. [81], which emphasizes transparency and authors’ reflectivity as essential components of conducting thematic analysis, a personal statement is provided in Appendix D.

## 3. Results

### 3.1. Survey Data Analysis

The total sample for this study consisted of 391 participants. None of the questions in the survey were mandatory and unanswered questions were treated as missing values and excluded in the analysis. The majority of participants had attained an undergraduate education level (36.6%), followed closely by those with a postgraduate degree (35.3%). Participants with a high school level degree made up 14.4%, and those with a PhD education level constituted 7.7%. In terms of age distribution, 32% of participants were between the ages of 25–34, followed by age groups 45–54 (22.8%), 35–44 (21.5%), above age 55 (20.2%), and 18–24 (3.4%). The gender distribution was predominantly female, with 96.2% of participants identifying as such, while 2.6% identified as male. Geographically, 26.3% of participants were from the USA and 24.0% were from Britain. Asian countries contributed 13%, European participants made up 6.6%, Canadians represented 6.9%, Australians accounted for 4.6%, and a mixed category comprised 2.8%. In terms of household composition, 34.5% of households had children and 65.2% did not. Regarding dog caregiving experience, 38.1% reported being experienced main caregivers for dogs, 24.8% had had family dogs before, and 24.3% were first-time dog caregivers.

Across the dataset, the highest mean values were observed for the authoritative received parenting style (authoritative: 2.99, authoritarian: 2.55, permissive: 2.19) and the authoritative dog-directed parenting style (authoritative: 4.25, permissive: 2.16, authoritarian: 1.74). Notably, protectionistic emerged as the predominant orientation toward dogs, attaining the highest mean score (protectionistic: 4.24, humanistic: 3.74, dominionistic: 2.58). Table 1 provides an overview of the demographic characteristics of the study sample and the participants’ previous dog caring experience.

### 3.2. Intergeneration Transmission Effect in Received Parenting Style and Dog-Directed Parenting Style

The results, as shown in Table 2, indicate that there is indeed an intergenerational transmission effect, specifically between received and dog-directed permissive parenting styles. Additionally, the results showed that a positive correlation between received permissive parenting style and dog-directed permissive parenting style, whereas a negative correlation was found between received authoritative parenting style and dog-directed permissive parenting style.

### 3.3. Stratified Analysis of Different Demographic Factors

When we stratified the data by demographic, the transmission between received permissive parenting style and dog-directed permissive parenting style remained, except in the Asian group. Additionally, a positive relationship was found between received authoritarian parenting style and dog-directed authoritarian parenting in the group that had previously had a family dog (Table 3).

### 3.4. Orientation toward Non-Human Animals Mediates Received Parenting Style and Dog-Directed Parenting Style

The potential mediating effects of orientation toward non-human animals in the relationship between received parenting style, dog-directed parenting style, and human–dog interaction were examined. The findings are presented in Figure 3, Figure 4, Figure 5 and Figure 6.

#### 3.4.1. The Effect of Orientation in the Relationship of Received Authoritarian Parenting Style and Dog-Directed Authoritative Parenting Style

Figure 3 illustrates that an increase in received authoritarian parenting style predicted higher scores in both humanistic and protectionistic orientations toward non-human animals. These orientations, in turn, positively related to dog-directed authoritative parenting style. The direct effects of received authoritarian parenting style and dog-directed authoritative parenting style are both positive and significant. Additionally, humanistic and protectionistic orientations mediate the relationship, with 36.40% of the effect mediated by humanistic orientation and 42.85% mediated by protectionistic orientation.

**Figure 3 animals-14-01038-f003:**
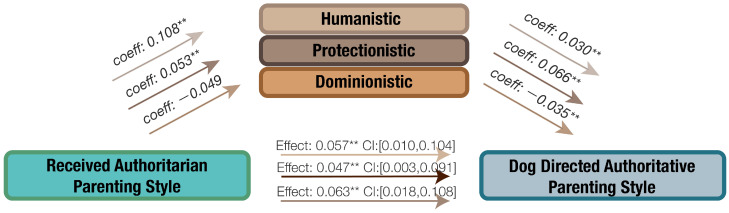
The mediating effect of orientation toward non-human animals on the relationship between received authoritarian parenting style and dog-directed authoritative parenting style, ** *p* < 0.01.

#### 3.4.2. The Influence of Orientation in the Relationship of Received Authoritarian Parenting Style and Dog-Directed Authoritarian Parenting Style

In Figure 4, no significant direct effect between received authoritarian parenting style and dog-directed authoritarian parenting style was observed as indicated by confidence intervals overlapping zero in the bottom arrow. Nonetheless, an increase in received authoritarian parenting style predicted an increase in protectionistic orientation toward animals, which in turn decreased dog-directed authoritarian parenting style. Bootstrapping analysis revealed a significant indirect effect on received authoritarian parenting style and dog-directed authoritarian parenting style (Effect: −0.0128, CI: [−0.0372, −0.0082]), indicating that a protectionistic orientation toward non-human animals might serve as a negative mediator between received authoritarian parenting style and dog-directed authoritarian parenting style.

**Figure 4 animals-14-01038-f004:**
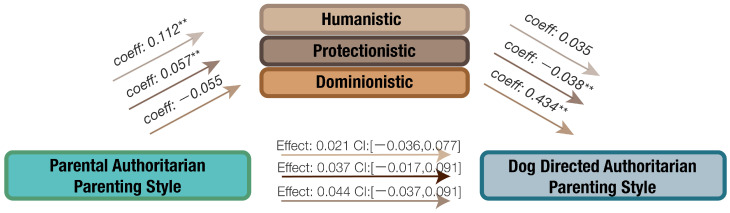
The mediating effects of orientation toward non-human animals on the relationship between received authoritarian parenting style and dog-directed authoritarian parenting style, ** *p* < 0.01.

#### 3.4.3. The Influence of Orientation in the Relation of Received Authoritarian Parenting Style and Dog-Directed Permissive Parenting Style

Similarly, no significant direct effect between received authoritarian parenting style and dog-directed permissive parenting style was found (Figure 5). However, received authoritarian parenting style positively predicted humanistic orientation, which in turn relates to dog-directed permissive parenting style. The proportion of the effect mediated by humanistic orientation was 68.29%. When protectionistic orientation was included as a mediator and controlled for, the relationship between received authoritarian parenting style and dog-directed permissive parenting style was no longer significant (Effect: 0.0639, CI: [−0.0002, 0.0128]). Bootstrapping analysis revealed a significant indirect effect linking received authoritarian parenting style to dog-directed permissive parenting style. This suggests that protectionistic orientation toward non-human animals might serve as a potential mediator in this relationship (Effect: 0.0126, CI: [0.0021, 0.272]).

**Figure 5 animals-14-01038-f005:**
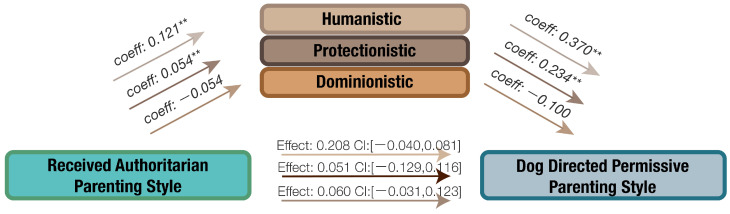
The mediating effects of orientation toward non-human animals on the relationship between received authoritarian parenting style and dog-directed permissive parenting style, ** *p* < 0.01.

#### 3.4.4. The Effect of Orientation in the Association of Received Permissive Parenting Style and Dog-Directed Permissive Parenting Style

Figure 6 indicates that an increase in received permissive parenting style led to higher scores in humanistic orientation toward non-human animals, which subsequently increased dog-directed permissive parenting style. The direct effect of received permissive parenting style on dog-directed permissive parenting style was positive and significant. The proportion of the effect mediated by humanistic orientation was 14.05% (Effect: 0.0433, CI: [0.0101, 0.0794]).

**Figure 6 animals-14-01038-f006:**
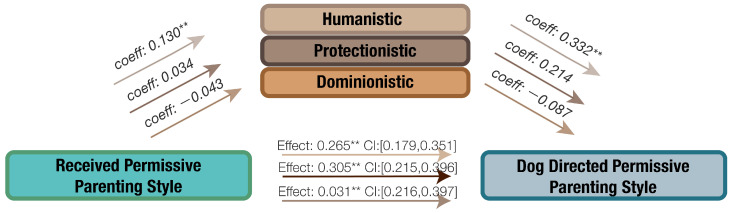
The mediating effect of orientation toward non-human animals on the relationship between received authoritarian parenting style and dog-directed permissive parenting style, ** *p* < 0.01.

The effects of orientation towards non-human animals on the remaining combinations of received parenting style and dog-directed parenting styles (permissive–authoritarian, permissive–authoritative, authoritative–authoritarian, authoritative–authoritative, and authoritative–permissive) did not show any significant relationships and are shown in Appendix E.

### 3.5. Qualitative Data Analysis

A total of 10 participants were randomly selected from those who indicated their willingness to participate in the interview (Table 4).

The analysis identified two overarching themes: theme (1), exploring the complex interplay of human–dog relationships: child, more than human, or just “Dog.”; theme (2) continuity, discontinuity, and compensation: self-reported influence of childhood parenting behaviors on dog-directed parenting behaviors. Table 5 presents a summary of each theme and subtheme along with their respective definitions.


*Theme 1 Exploring the Complex Interplay of Human–Dog Relationships: Child, More Than Human, or just “Dog.”*


This theme delves into the complex role dogs play in our relationship, intricate and changing nature of the human–dog relationship, revealing a dynamic parenting role assumed by caregivers within the context of the more-than-human concept, highlighting how caregivers view their dogs not merely as pets but as integral members of the family, reflecting the intertwining of human and canine lives within a shared social group. Furthermore, it explores the underlying reasons behind the multifaceted roles that dogs play in these relationships.


*Subtheme 1A: Blurring the Human–*
*Dog and Parent–*
*Child Relationship Boundary: Exploring Family Construction, Recourse Dominance, and Dogs’ Transitions into Child-Like Roles*


Participants drew parallels between their interactions with dogs and parent–child relationships, emphasizing the significance of comprehending the unique needs and temperaments of dogs. Frequent references to terms such as “companion”, “family”, and “friends” when discussing the human–dog relationship demonstrate the depth of emotional connection and kinship-like practices. Despite not explicitly identifying themselves as “parents”, caregivers share similar narratives in their interactions with both human children and dogs, reflecting the complexity of the human–dog bond. For instance: 


*“I try to understand their [dog] individual needs and their [dog]individual temperament and urm how they [dog] like to live their [dog] life ….. I try to let them [kid]make their own adjustment and possible try to do something similar where I would allow freedom to be there in a support role—participant 02”*



*“We [the dog and the participant] are like family, I guess, like family but. Families and friends, we companion each other, and we love each other.—participant 01”*


Similarly to parent–child relationships, in which parenting behaviors change to correspond to different phases of child development [82], participants noticed that as their dogs grew and transitioned through different life stages, the nature of their relationship and interactions evolved accordingly to align with the dog’s changing condition.


*“I mean it’s shifted a lot as he gotten older. I think he’s the first dog I’ve gotten in a long time that’s really forced me to learn new things and sort of think differently about the way I interact with him. Urm he now is a fairly manageable pet in many situations but that was certainly not true until the last three couple of years he started to really be affected by age.—participant 05”*



*“Urm so I have adjusted, I I don’t walk my dogs together, I walk them separately. So he [the older dog] goes for shorter, slower walks and then. Urm we don’t really do any training with him [the older dog] anymore. He’s still enjoys doing a bit of tracking, so I do try and do tracking with him [the older dog] quite regularly and then I work the other dog my Malinois separately, he goes for longer walks and we do more training and more activities.—participant 02”*


The construction of the family unit emerges as a key factor influencing how caregivers position dogs within a role akin to that of a child. This familial relationship structure, marked by dogs’ subordination to their caregivers, often brings dogs closer to children within the family dynamics. Due to their dependency on caregivers for access to essential resources, dogs are placed in a position of unequal power, wherein they are perceived as needing care and support, casting dogs from the perspective of “needing to be taken care of.” 


*“I would say urm [my relationship with dog] very close cause I take care of him almost every day.–participant 03”*



*“We had to go to work in the morning. So we usually would expect to walk the dog early in the morning, like maybe around 6:30 or something like that…So walk the dog in the morning at 6:30, then we might have dinner at around 7:00 o’clock and that’s when my dog had his dinner.—participant 07”*


Additionally, the formation of two-person households contributes to the perception of dogs as substitute children. This contrasts with Bouma et al. [83] study on cats’ caregivers, which suggested caregivers who perceive their cat as a child or friend usually live alone, whereas caregivers who do not live alone tend to view their cat as “just a pet”. Research indicates potential distinctions based on the pet’s species. Dogs tend to provide a stronger mental and emotional support, forging close attachment compared to other species [84,85,86]. Carr and Pendry [87] discovered that students who preferred dogs as their favorite non-human animals exhibit a higher degree of human substitution compared to students who preferred cats.


*“He thinks more of a pet owner than a parent kind of role. I think that’s because he is very used to having a dog in his household, so it’s more like a partner or while in his family, his family, his whole family would take care of the dog, so it’s more like a companion. But for us, we are only a couple. So two people with one dog. So we have to be sort of like parents for our dogs.—participant 03”*



*Subtheme 1B: More-than-Human Practices: Insights from The Increased Knowledge of Dog and Experienced Childhood Loneliness*


Relationships with dogs were not always characterized by viewing them as “almost human,” but rather as “more family than family.” In some cases, dogs were valued more highly than humans and caregivers perceived a stronger attachment to their dogs than to certain individuals within their human family [88]: 


*“I’d rather have a dog than a child being perfectly honest.—participant 08”*



*“He knew for example that if he said he didn’t want the dog anymore that he didn’t like the dog or the dog he he didn’t want the dog in the bedroom, he knew that he would be out the bedroom, not the dog. That’s the way it was.—participant 09”*


The term “training” is consistently emphasized throughout the narratives. Collaborating with professionals or pursuing relevant courses equips individuals with essential knowledge about a dog’s agency and species-specific needs. This understanding becomes a crucial factor when individuals seek to incorporate dogs into their everyday routines. As Irvine [89] noted, the curiosity and willingness to learn about another being’s mind produce a different relationship that requires commitment, compared to having a pet just for pleasure. 


*“It’s really important to me that I like my dog has agency and is urm able to express their natural behaviors but it’s also equally important to me that my dog is safe for society and those around me and myself, and balancing those to things, especially for the breed that I got can be a little of a trick. Urm I definitely puts my dog’s quality of life first, I’m not going to sort of prioritize long term anyway, my own needs over their needs.—participant 05”*



*“I spend a lot of time thinking about behavior and working with people who are struggling with their dog’s behavior and umm and I have intentionally like I intentionally got a very high energy high drive dog with probably some, you know, behavioral quirks that would make him difficult in the average home because that’s the kind of dog that suits my life well. And so my my professional work definitely influences our interactions because I know a lot more that now than I did when I was not professionally training.—participant 10”*


The increasing knowledge about dogs as a species prompts participants to make conscious efforts to avoid excessive anthropomorphism, recognizing its potential detriment to non-human animal welfare. This caution arises from the challenge of distinguishing their own needs and emotions from those of their dogs, despite acknowledging that their canine companions sometimes fulfill a role akin to a “replacement child.” This finding aligned with previous research that had also identified similar narratives where individuals viewed their non-human animals as surrogate children while making conscious efforts to resist anthropomorphism. The effort to resist anthropomorphism reflected an awareness of the boundaries and differences between humans and dogs, while still valuing the deep emotional connection and caregiving role they have with their pets [19]. 


*“I try to understand their individual needs and their individual temperament and urm how they like to live their life….. And then fortunately with my Belgian shepherd, he did become what I wanted him to become, and that I really I I let my dogs live live their own lives….. They live with me in the house, they. Urm they they’re basically like. I like to treat them like dogs, but they they’re a little bit like children in in my house.—participant 02”*


When discussing routines in the home, most participants reported a lack of a structured routine for their dogs, with activities being performed at specific times considering both humans’ and dogs’ needs rather than following a consistent schedule. Power [8] emphasizes the expansion of the family concept, as caregivers intentionally integrate dogs as dogs into their daily lives. The strong bond between humans and dogs is fostered through shared living spaces and daily interactions, nurturing a deep understanding of each other’s needs. Moreover, several caregivers mentioned prioritizing their dog’s needs and adjusting their own lifestyle accordingly. As proposed by Barlow et al. [90], women who had experienced childhood neglect tended to develop stronger attachments to companion animals compared to others. In the context of this study, the inclination to prioritize a dog’s needs was also associated with a higher frequency of being left alone during childhood. This highlights a potential connection between early experiences and the nature of human–dog relationships.


*“I think because I was a lot on my own, as I said, not not in the house on my own, but I was left to play on my own because obviously my dad was at work. [….] So I think I got used to being very independent and I play on me and I play with dolls on me and teddy bears starts off with just love playing with teddy bears and then start like that sort of passed on to animals……Everything is geared up to them. Really, I would never. For example, I would never come down and make my breakfast and say ohh you know they can wait for a while for theirs. It’s all about getting them comfortable first because I knew I wouldn’t be wanna be left waiting for the toilet. You know, they they rely on me to walk and look after them and I wouldn’t want to be bursting to go to the loo while somebody was waiting making a cup of tea. So I I just everything’s about their point of view.—participant 09”*



*Subtheme 1C: Situations Where a Clear Boundary Between Human–Dog and Parent–Child Relationships Exists: When Human or Dog can’t Meet One’s Own Expectation*


Nonetheless, some participants highlighted the differences between parent–child relationship and human–dog relationships. One distinguishing factor between the parent–child relationship and the human–dog relationship was the acknowledgement of the shorter lifespan of dogs. Participants in the study recognized that the limited lifespan of dogs sets their relationship apart from that of parents and children. This awareness prompted them to discontinue using parenting approaches similar to those employed in parent–child relationships. As a result, there were reduced expectations and a shift towards prioritizing the dog’s immediate wellbeing and quality of life, rather than projecting long-term aspirations and goals, as is often the case in parent–child relationships.


*“So we just want him to be a happy dog enjoying his well short life so we didn’t expect a lot from him and. Certainly we don’t, we don’t think he would have achieved anything in the future. So we just let him be what he is urm as he his own personality and ways. And basically that differs a lot from how I was treated by my parents.—participant 03”*


One participant who rejected the label of being a pet parent referred to herself as a “sister”, she stated the following:


*“Most of the fur parents they would call, they are the parents of the dog most most I. But that’s never how I look myself with my dog. I don’t I never think of me being a parent. I don’t think I’m a friend. I’m. I think that I’m. I always call that I am ... Yeah, I’m her sister….. I wasn’t prepared to accept that six years, six or seven years ago, I was just I don’t know. I was just 20, 22, I guess I was just 22 years old. I could barely feed myself….. I think I avoided being a parent at that time. But now I still avoid being a parent because if I think of myself being parent and I have to do more effort. So yeah, maybe that’s why I I keep avoiding that because I don’t want to do more effort right now.—participant 06”*


Despite participants rejecting the label of being a dog parent and excessive anthropomorphism, the study still identified narratives of kinship practices, aligning with Charles [19] suggestion that dogs and humans are seen as part of the same social group or family. Previous studies have described the practice of maintaining distance between non-human animals and humans, treating them as “just animals,” which allows for a more objectifying perspective or an egalitarian viewpoint from animal activists [21]. However, the findings of this study suggest that there may be more complex factors at play. The concept of “human-parenting,” which entails responsibility, effort, and expectations, can also shape how people construct their roles and interactions with dogs. In a manner akin to family systems, where parents’ roles are characterized as complex due to the necessity of attending and responding to the needs of multiple family subsystems, a high level of caregiving burden in interspecies families may lead to self-sacrifice and subsequently impact responsive parenting or lead to withdrawal [91,92]. This pattern is also evident in pet parenting, where individuals view their pets as integral family members. These findings highlight the complexity of human–pet relationships, resonating with the “more-than-human” model, which goes beyond a simple pet–caregiver dynamic and emphasizes the significance of kinship practices in fostering improved canine welfare and strengthening bonds between humans and their canine companions.


*Theme 2 Continuity, Discontinuity, and Compensation: Self-Reported Influence of Childhood Parenting Behaviours on Dog-Directed Parenting Behaviour*


This theme encapsulates the cross-species intergeneration continuity and discontinuity in parenting behavior. It also delves into participants’ reflections on how their experiences with parenting and the human–dog relationship may serve as a form of compensation for their childhood experiences.


*Subtheme 2A: Continuity of parenting behaviors and attitudes with high responsiveness*


Participants who reported sharing a positive relationship with parents and perceived received parenting behaviors as supportive tended to use a similar parenting approach. This result echoes with Savelieva et al.’s [93] suggestion that receiving higher degrees of warmth and acceptance in the parent–child relationship is a potential predictor of replicating that positive parent–child relationship.


*“urm bringing him (dog) up like my mom and dad brought me, brought me up, which sounds a bit barmy, really, but. But, you know, they were generally just used to stay calm. I didn’t get shouted at a lot or anything because you know when people are nice to you and treat you nicely, you want to reciprocate and then this.—participant 08”*



*My view that my parents treated me as kind of an independent human being and not like a little robot that was an extension of themselves is very much how I view my dog and also how I anticipate viewing my children.—participant 10*


Moreover, it was not only positive parenting behaviors that showed intergenerational continuity, as participants who had experienced affectionate and positive attitudes from their parents toward non-human animals generally adopted similar forms of behavior towards their dogs. This aligns with prior research suggesting that parental attitudes and behaviors regarding external entities can influence subsequent generations, even when these behaviors are not directed towards the individual themselves [94]. 


*“I would always have responded like this to animals that need safety or show distress urm and I think that is more from the way that my parents uh dealt with animals and how our family always had animals and were very conscious and aware of animals around us from when I grew up until I lived home.—participant 02”*



*Subtheme 2B: Discontinuity of High Control Parenting Behaviors and Attitudes*


On the other hand, participants who described sharing a stricter relationship with their parents and recalled negative emotions associated with their childhood rejected similar parenting patterns when interacting with dogs. One participant who described her relationship with parents as “a disaster and like hell” changed her interaction as soon as she realized that she had adopted a similar parenting behavior to that of her parents:


*“I literally don’t want to be the kind of parents which my parents they are so…. I think about one thing is like my parents want me to do lots of things they want. So like in the in the beginning when I and with my dog in the beginning, I guess I am a little bit like that, like I wish my dog can be talented, something like that. Like I want to do the dog dance dog dancing sport with her, and that’s my dream. But I just put my dream, like on the dog. I guess it’s a little bit like when my parents did to me. But when I realized that, after I realized that I try my best to do things differently. Like I I will try to be more understanding for the dogs. Like. Umm, get to know what she really wants and what she really likes to do and. Try new things with her. Just don’t focus on the things I want.—participant 01”*


In addition, when participants perceived that their experience with their family dogs in childhood did not align to their attitudes toward dogs in adulthood, a discontinuity of parenting style occurred. This discrepancy between attitudes was justified by the participants by explaining the different concept of dog caregiving practices during their childhood compared to the present. They recognized that societal views and cultural values regarding dog caregiving practices have changed over time, leading to changes in their own parenting styles towards dogs. This is consistent with Littlewood [51]‘s suggestion that societal views and trends in parenting styles played an important role in shaping the discontinuity of parenting practices. 


*“I feel like it’s easier for me to draw a line between the way I saw my parents interact with our dog growing up and how I interact with my dog then it’s around my relationship around with my parents with how I interact with my do......so I think that I very deliberately have chosen not to live with my dog that ways when I saw my parents with our first dog if that makes sense. And to be fair to them, like they too since them right, like they also would have choose not to do that again but it was certainly what they knew at the time.—participant 05”*



*Subtheme 2C: Compensating for Low-Responsive Parenting Behaviors and Attitudes*


An alternate of differential intergeneration parenting is compensation in parenting behaviors, particularly if the participant perceived the parenting they received as less than ideal. Consistent with the compensation hypothesis [95], which suggests that individuals may seek to compensate for their own distant relationship and deficiencies in parenting from their family of origin by displaying greater affection and care towards their offspring, participants reported that they were more inclined to show affectionate behaviors to their dogs [95,96]. Compensating parenting behaviors continued even when the behaviors were not beneficial in human–dog relationship. Participants reported feeling that they were spoiling the dog and putting dogs’ needs in front of other family members and their own needs in order to avoid letting the dog down.


*“With my experience as a child with my parents, would make me more loving and more caring. Sometimes could spoil my dog, I would say. And also I kind of, I kind of feel like there are things I would, like things I would do to make my dog happy. For example, snacks or taking him outside more frequently or petted more frequently, things like that, or just basically no disciplines at all.—participant 07”*



*“I think because I was a lot on my own …. So I think I got used to being very independent and I play on me and I play with dolls on me and teddy bears starts off with just love playing with teddy bears and then start like that source of passed on to animals… with animals, just think you’ve got absolutely no voice that they’re completely are mercy animals. And I’ve always hated and really struggled with any form of animal cruelty. They’re completely they rely on us for food. They rely on us not to hurt them and they’ll let down all the time…[My daily life] Everything is geared up to them [dog]).—participant 09”*


Notably, despite participants demonstrating awareness of the potential influence of the parenting they received and their childhood experiences on their dog-directed parenting behaviors, when focusing on their dog-directed parenting behaviors in specific scenarios related to responsiveness (comforting under stress), rules and routines (demandingness), and exploration, they did not directly report the influence of their own received parenting behaviors. Instead, they emphasized their perception of certain parenting behaviors that they believed to be important or beneficial for their dogs, and subsequently chose to adopt them. Consequently, regardless of participants’ perceptions of their parents’ parenting behavior during childhood, whether authoritarian, imposing limited autonomy, and withholding emotions, or encouraging, supportive, and comforting, all participants exhibited similar attitudes towards encouraging exploration, providing enrichment, and showing willingness to comfort their dogs when they displayed distress behaviors.


*“[….] not really didn’t I supposedly they [parents] didn’t really like it.... They [parents] were both are supposed quite protective of me. …. I’m into in anything that makes their [dog) life better. I am keen to let them [dog] try things.—participant 09”*


The findings suggest that the emotions experienced in response to certain parenting behaviors can influence participants’ awareness and motivation to either continue, discontinue, or compensate for those behaviors when interacting with their dogs. However, when implementing other parenting patterns that do not evoke strong emotional responses, participants’ awareness of the influence from their own received parenting style may diminish. In such instances, they may attribute their decision to adopting specific parenting practices for their dogs to reasons other rather than drawing connections to their own upbringing or childhood experiences.

## 4. Discussion

Overall, we found that parenting style shows complex cross-species, intergenerational transmission dynamics, which can be mediated by attitudes towards non-human animals. Moreover, the quantitative and qualitative approaches were highly complementary with the qualitative data shedding light into the thought processes that participants go through when making parenting-style choices that lead to the dynamics we found in the quantitative data. The qualitative data also uncovered the emergence of more-than-human kinship practices in multi-species families.

The first aim was to investigate intergenerational transmission effects in multispecies family systems. The quantitative findings demonstrated that a permissive parenting style exhibited this transmission effect between received parenting style and dog-directed parenting style, whereas a negative correlation was observed between received authoritative parenting style and permissive dog-directed parenting style. This aligns with Campbell and Gilmore’s [49] earlier findings, which suggested that authoritative parenting style does not show a transmission effect, although it is considered a more optimal parenting style. The lack of continuity in this context may be attributed to the different perceptions between parents and their children when evaluating self and received parenting styles. Moreover, the continuous shifting social–cultural orientation toward child rearing might also impact the nature and perception of authoritative parenting, which is also consistent with Ghorbani et al.’s [97] suggestion that the changing view in the society can lead to the discontinuity of parenting styles between generations. 

Our study identified several variations in the intergenerational transmission effect across nationality and dog caring experience. This suggests that culture and received attitude play a role in influencing the transmission of parenting styles between generations. Notably, Asian participants in the qualitative research revealed a propensity to discontinue their received parenting behaviors, aligning with the lack of a significant transmission effect in the stratified analysis. Globalization and greater influence from Western culture have shifted Asian parenting styles toward an authoritative approach, suggesting one of the reasons for the discontinuity [98]. Emphasizing individual wellbeing is accompanied by a parallel rise in individuality. Sun and Ryder [99] proposed that Chinese parents have an increased tendency to adopt a more affectionate-based interaction with the emergence of individualism. Echoing Littlewood [51], sociocultural shifts in values, attitudes, and behaviors across generations could impact parenting behaviors. 

The results of the second aim demonstrated that orientation toward dogs plays a crucial mediating role in the transmission of parenting style. Ellingsen et al. [100] emphasized that caregivers’ attitudes toward dogs are correlated with their attachment to dogs and their perception of pain in dogs, which could significantly impact the welfare of dog. Similarly, caregivers who perceive their cats as their children or best friends tend to view cats as more empathetic and dependent on them, leading to increased involvement in their care and a higher likelihood of having multiple cats [83]. These findings align with previous research and underscore the significant impact of orientation toward non-human animals on parenting styles. Moreover, humanistic orientation and protectionistic attitudes toward non-human animals played a critical mediating role in the discontinuity of received authoritarian parenting style. Protectionism decreased the likelihood of the transmission of authoritarian parenting style and increased the tendency to adopt dog-directed authoritative or permissive parenting styles. Humanistic orientation also showed a similar effect by increasing the tendency to adopt dog-directed authoritative or permissive parenting styles. In addition, humanistic orientation serves as a mediator amplifying the intergenerational transmission effect between received and dog-directed permissive parenting styles. These findings highlight the importance of caregivers’ attitudes toward non-human animals in shaping parenting behaviors and human–dog interactions. 

The qualitative data revealed the emergence of more-than-human kinship practices, accentuating the active involvement and cohabitation of both humans and dogs in everyday life. Caregivers demonstrated a perspective beyond viewing their dogs merely as replacements for children, as they acknowledged the dogs’ perspectives and agency as dogs, enabling the dogs to actively partake in daily routines. This observation aligned with Volsche [101], who observed a dynamic fluidity in the human–dog relationship, wherein caregivers negotiate their dogs’ agency while simultaneously acknowledging their dependency. One explanation for the increased acknowledgment of a dog’s dependency and agency is the growing popularity of positive training. Positive training, which emphasizes a “dog-centric approach,” encourages caregivers to “see the world from a dog’s point of view” and consider the dog’s individual temperament and preferences, changing how people view their dog from a “replacement child” to another respected living being in the relationship [102,103]. This aligns with Greenebaum’s [104] research, indicating that a dog-centric training approach influences perceptions of the roles and status of both the dog and the caregiver. This mutual understanding fosters strong bonds and facilitates identity management within the relationship [101]. This bond is further strengthened by the seamless integration of dogs into the family’s daily life, surpassing solely planned interactions.

Moreover, caregivers’ awareness of dogs’ agency and their efforts to avoid anthropomorphism, reflected a protectionistic orientation toward non-human animals and underscored the significance of considering orientation toward dogs in human–dog interactions. These findings align with the quantitative data, reinforcing the importance of considering caregivers’ attitudes and orientations toward non-human animals in comprehending and promoting positive human–dog interactions. Consistent with the quantitative findings on intergenerational transmission, our qualitative analysis also identified continuity in parenting behaviors, particularly when caregivers perceived positive and affectionate parenting behaviors from their parents. This continuity of affectionate and positive attitudes aligns with the intergenerational transmission effect observed in permissive parenting style, characterized by high responsiveness, affection, and sensitivity to others’ needs. 

Furthermore, childhood experiences and attitudes play a critical role, as demonstrated by Meeusen [105], who found a transmission of environmental attitudes within families. Gauly [106] also reported a significant transmission of reciprocity between generations. In our study, participants reported that parents who demonstrated empathy and affinity toward animals had a positive impact, enabling the adoption of similar perspectives and relationships with dogs since childhood.

In contrast, when caregivers perceived themselves sharing a stricter relationship with parents and the associated negative emotion in childhood, they consciously rejected practicing similar parenting behaviors. This rejection may underlie the discontinuity between received authoritarian parenting style and dog-directed authoritarian parenting style, which is characterized by high demandingness and low responsiveness. This discontinuity was mediated by protectionist attitudes towards non-human animals. Protectionism had a suppressive effect, in that it reduced the likelihood of someone who experienced authoritarian parenting exhibiting authoritarian parenting with their dog (Figure 4). Another factor contributing to discontinuity in parenting patterns is a discrepancy between participants’ childhood experiences with family dogs and their current orientation toward dogs. This discrepancy is often justified by the changing societal values of dog parenting over time. Recent studies have highlighted a significant reduction in the use of physical reprimands by caregivers, indicating an increased awareness of companion animal welfare [107]. This shift underscores the acknowledgment that positive punishment may jeopardize the human–dog bond and exacerbate to the occurrence of undesirable behaviors [108]. As presented by Littlewood [51], the increase in educational levels and sociocultural shift could contribute to the discontinuity of parenting styles across generations. Despite caregivers’ reported efforts to deliberately choose different parenting behaviors, the intergenerational transmission effect persisted in the group that had a family dog during childhood when we stratified the quantitative data by dog-caring experiences. This suggests that, despite being aware and making efforts to discontinue high demandingness in parenting behavior, their childhood experiences might unconsciously influence some of their current parenting behaviors.

Moreover, compensation effects emerged in the qualitative analysis, where caregivers displayed increased affection and care for their pets when they perceived their experiences of being parented as less ideal. This aligns with Zilcha-Mano et al.’s [109] recognition of the tendency for individuals to project their previous experiences and feelings onto therapy dogs, creating a compensating relationship to fulfill unmet attachment needs. Similarly, in our study, caregivers projected their childhood experiences and emotions onto their pets, engaging in behaviors aimed at preventing the same negative experiences from happening to their dogs. This projection of unmet needs in childhood might also explain the positive relationship between received authoritarian parenting style and humanistic orientation toward non-human animals, which served as a significant mediator for a permissive parenting style. Individuals with a higher level of authoritarian parenting style tended to exhibit an increased tendency toward avoidance attachment, which has been reported to be associated with a greater likelihood of viewing their animal as a substitute for human companionship [110,111]. In this study, participants who experienced authoritarian parenting and held humanistic views compensated by directing permissive parenting towards their dogs (Figure 5). However, it is important to note that the projection and displacement of one’s own experiences and needs may hinder caregivers’ abilities to recognize and address other’s needs and abilities, potentially leading to an inadequate and overprotective parenting style [55]. 

The findings from this study point to several potential directions for future research. First, the participants were predominantly female, possibly because the term “parenting behaviour” used in title of the participant recruitment materials may have been more appealing to women, who are more likely to identify themselves as “pet parents” [21]. Previous research has suggested gender differences in human–dog interaction and attitudes toward non-human beings [112,113]. Women have been observed to exhibit an increased tendency towards verbal behaviors, with their utterances often resembling infant-directed speech or “motherese” [114]. Future research should strive to include a more balanced representation of male participants. Secondly, inclusion of higher numbers of participants across multiple cultures would increase our understanding of how cultural background may influence cross-species transmission of parenting behaviors. 

Furthermore, the exclusion of participants with mental health issues limits our understanding of the correlation of dog-directed parenting styles and mental health. Since abundant research has identified the significant impact of human–dog relationships on mental health, there is a distinct need to include mental health measurements to allow greater understanding of the effects on dog-directed parenting styles. Appropriate measurements assessing the correlation between various mental health states, dog-directed parenting styles and intergenerational transmission are critical to broadening our understanding of human–dog relationships. Finally, the results shed light on the relationship between orientation toward animals and parenting behavior. Further research could delve into the association between orientation toward dogs and the choice of training method, as previous studies have suggested a correlation between training method and the occurrence of undesirable behaviors in dogs [108]. This suggests that the adoption of training methods may be closely related to canine welfare.

## 5. Conclusions

In conclusion, our study provides valuable insights into human–dog interactions by showing that parenting styles have complex transmission dynamics that occur across species in multi-species kinship practices. The quantitative and qualitative data were highly complementary, with the qualitative data showing that some of the broader statistical patterns are likely the effects of conscious choices that participants are making. For example, we detected cross-species, intergenerational transmission effects for permissive parenting in that participants who experienced permissive parenting were likely to use it with their dogs and during interviews, participants indicated that they were deliberately replicating their own experiences of receiving responsive parenting. Similarly, orientation toward non-human animals played a crucial mediating role between received parenting style and dog-directed parenting style in that protectionistic attitudes reduced the likelihood of participants who experienced authoritarian parenting replicating that with their dogs and a humanistic attitude increased the likelihood that they would compensate for an authoritarian upbringing by being permissive with their dogs. These processes were also supported by the comments that participants made during interviews by emphasizing the role of perceived childhood experiences and emotions in influencing people’s awareness of adopting specific parenting behaviors. Our research contributes to the expanding knowledge of the multifaceted nature of human–dog interactions and establishes a strong foundation for future investigations exploring the interplay between parenting styles, cultural influences, and the wellbeing of our canine companions. Additionally, by recognizing the impact of parenting styles on dog-directed behaviors, our findings offer practical implications for dog training and behavior modification programs, emphasizing the need for tailored approaches that account for individuals’ upbringing and orientations toward non-human animals. Such insights can ultimately contribute to more effective strategies for enhancing welfare and for fostering positive relationships between humans and dogs in real-world settings.

## Figures and Tables

**Figure 1 animals-14-01038-f001:**
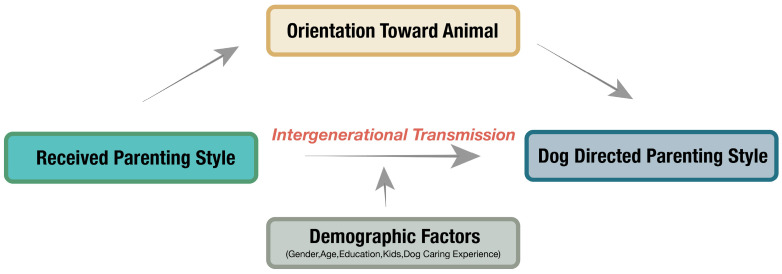
The core concept of this study.

**Figure 2 animals-14-01038-f002:**
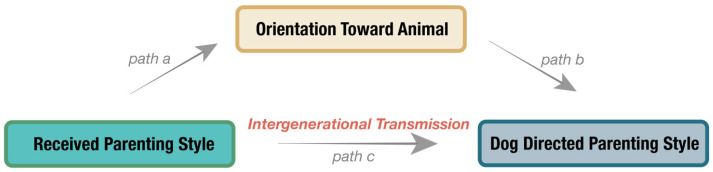
Overall mediation model.

**Table 1 animals-14-01038-t001:** Demographic Statistics for participants in quantitative study.

Variable		N	Authoritarian		Authoritative		Permissive		Dog-Directed Authoritarian		Dog-Directed Authoritative		Dog-Directed Permissive		Protectionistic		Dominionistic		Humanistic	
			$\overset{-}{x}$	SD	$\overset{-}{x}$	SD	$\overset{-}{x}$	SD	$\overset{-}{x}$	SD	$\overset{-}{x}$	SD	$\overset{-}{x}$	SD	$\overset{-}{x}$	SD	$\overset{-}{x}$	SD	$\overset{-}{x}$	SD
Age	18–24	14	3	1.3	2.8	1.1	2.2	0.8	1.6	0.4	4.5	0.3	2.2	0.5	4.4	0.2	2.3	0.6	4	0.5
	25–34	125	2.7	0.9	3	0.9	2.3	0.5	1.9	0.4	4.3	0.3	2.3	0.5	4.3	0.3	2.6	0.5	3.9	0.5
	35–44	84	2.5	0.8	3	0.8	2.2	0.6	1.7	0.5	4.3	0.5	2.2	0.5	4.3	0.3	2.4	0.6	3.9	0.5
	45–54	89	2.5	0.8	2.9	0.9	2.1	0.6	1.7	0.5	4.1	0.4	2.1	0.5	4.2	0.3	2.7	0.6	3.6	0.5
	55+	79	2.4	0.9	3	0.9	2.1	0.6	1.6	0.4	4.2	0.5	2	0.6	4.2	0.4	2.7	0.5	3.4	0.5
Gender	male	10	2.2	0.5	3.6	0.7	2.2	0.5	2	0.5	4	0.5	2.1	0.6	4.1	0.4	2.7	0.7	3.7	0.8
	female	376	2.6	0.9	3	0.9	2.1	0.6	1.7	0.5	4.2	0.4	2.1	0.5	4.2	0.3	2.6	0.6	3.7	0.5
	NA	1	1.6		3.9		2.4		2.1		3.3		2.1		3.5		3.5		2.8	
	not applicable	1	4.2		1.7		1		1.1		4.9		2.5		4.4		1.9		4.6	
	other	3	2.8	1.2	2.4	0.6	1.9	0.8	1.6	0.2	4.1	0.2	2.3	0.4	4.3	0	2.6	0.2	3.9	0.4
Nationality	American	103	2.5	0.8	3	0.9	2.2	0.6	1.7	0.4	4.2	0.4	2.1	0.6	4.2	0.3	2.6	0.5	3.7	0.5
	Asian	51	2.6	1	2.9	0.8	2.2	0.5	2	0.5	4.1	0.5	2.4	0.5	4.1	0.3	2.7	0.5	3.9	0.4
	British	94	2.6	0.9	3.1	0.9	2.1	0.5	1.6	0.5	4.3	0.4	2.1	0.5	4.3	0.3	2.6	0.5	3.7	0.5
	European	26	2.4	0.8	3.2	0.8	2.5	0.5	1.8	0.5	4.3	0.5	2.3	0.6	4.3	0.4	2.5	0.6	3.6	0.6
	Canadian	27	2.5	0.9	2.9	0.8	2.3	0.6	1.7	0.4	4.3	0.3	2.1	0.6	4.2	0.5	2.6	0.4	3.6	0.7
	Australian	18	2.2	0.6	3.6	0.7	2	0.5	1.8	0.6	4.1	0.4	1.9	0.6	4.1	0.4	2.7	0.5	3.3	0.5
	Mix	11	2.5	0.9	3.1	0.9	2.3	0.8	1.9	0.7	4.2	0.5	1.9	0.5	4.1	0.4	2.8	0.6	3.7	0.5
	Others	7	2	0.7	3.3	1.2	1.9	0.8	1.4	0.4	4.5	0.3	2.2	0.5	4.6	0.2	1.8	0.5	3.7	0.5
	Missing	54																		
Kids in family	No	255	2.6	0.9	3	0.9	2.3	0.6	1.7	0.5	4.3	0.4	2.2	0.5	4.3	0.3	2.5	0.5	3.8	0.5
	Yes	135	2.5	0.9	3	1	2.1	0.6	1.7	0.5	4.2	0.5	2	0.5	4.2	0.4	2.8	0.5	3.6	0.6
Previous dog caring experience	Experienced caregiver (Family dog)	97	2.5	0.9	3	0.8	2.1	0.6	1.8	0.5	4.2	0.5	2.2	0.5	4.2	0.3	2.7	0.6	3.6	0.5
	Experienced caregiver (main caregiver)	149	2.6	0.9	2.9	0.9	2.2	0.6	1.7	0.5	4.3	0.4	2.1	0.5	4.2	0.4	2.5	0.5	3.7	0.6
	First time caregiver	95	2.6	1	3	0.9	2.2	0.6	1.7	0.4	4.3	0.4	2.2	0,5	4.2	0.3	2.6	0.6	3.7	0.5
Education Level	high school	57	2.7	1	2.8	0.9	2.2	0.6	1.6	0.4	4.3	0.4	2.2	0.6	4.3	0.3	2.5	0.6	3.7	0.5
	undergrad	143	2.5	0.9	3	0.9	2.2	0.6	1.7	0.5	4.3	0.4	2.1	0.5	4.2	0.3	2.6	0.5	3.8	0.6
	postgrad	138	2.5	0.9	3.1	0.9	2.2	0.6	1.8	0.5	4.2	0.4	2.2	0.6	4.2	0.3	2.6	0.6	3.7	0.6
	PhD	30	2.6	0.8	3	0.8	2.3	0.6	1.8	0.5	4.3	0.5	2.2	0.5	4.3	0.4	2.5	0.6	3.7	0.5
	Missing	23																		

Note: N denotes the number of participants in each category, and the values represent the mean of each indicator for the respective category; SD indicates the standard deviation.

**Table 2 animals-14-01038-t002:** Regression coefficients from regression predicting the impact of intergenerational transmission between received parenting style and dog-directed parenting style.

	Dog-Directed Authoritative Parenting Style					Dog-Directed Authoritarian Parenting Style					Dog-Directed Permissive Parenting Style				
	B	SE B	ß	t	*p*	B	SE B	ß	t	*p*	B	SE B	ß	t	*p*
Authoritative	0.01	0.04	0.02	0.27	0.79	0.05	0.04	0.10	1.19	0.24	−0.10	0.05	−0.16	−2.15	0.03 *
Authoritarian	0.05	0.04	0.09	1.11	0.27	0.08	0.05	0.15	1.85	0.07	−0.02	0.05	−0.03	−0.43	0.67
Permissive	−0.01	0.05	−0.01	−0.14	0.89	0.07	0.06	0.08	1.19	0.23	0.33	0.06	0.33	5.16	0.00 **

Note B = unstandardized regression coefficients; SE B = unstandardized regression coefficients standard error; ß = standardized regression coefficients. * *p* < 0.05; ** *p* < 0.01.

**Table 3 animals-14-01038-t003:** Multivariate linear regression examines the correlation of parenting styles stratified by demographic factors. Note that some categories with low numbers of participants (i.e., men) were not included or, where possible, they were combined (i.e., Asian nationalities).

			Dog-Directed Authoritative Parenting					Dog-Directed Authoritarian Parenting					Dog-Directed Permissive Parenting				
			B	SE B	ß	t	*p*	B	SE B	ß	t	*p*	B	SE B	ß	t	*p*
Age	18–44	Authoritative	0.00	0.06	0.00	−0.01	0.99	0.11	0.07	0.19	1.63	0.11	−0.10	0.07	−0.16	−1.44	0.15
		Authoritarian	0.06	0.06	0.12	1.00	0.32	0.13	0.06	0.23	2.05	0.04	−0.03	0.07	−0.05	−0.40	0.69
		Permissive	0.00	0.07	0.00	−0.02	0.99	0.03	0.08	0.04	0.39	0.70	0.31	0.09	0.32	3.55	0.00 **
	44–55	Authoritative	0.05	0.06	0.11	0.89	0.38	−0.01	0.06	−0.02	−0.17	0.87	−0.09	0.07	−0.15	−1.30	0.20
		Authoritarian	0.04	0.06	0.08	0.68	0.50	0.05	0.07	0.09	0.74	0.46	0.01	0.07	0.01	0.10	0.92
		Permissive	−0.03	0.08	−0.04	−0.37	0.72	0.09	0.08	0.11	1.11	0.27	0.34	0.10	0.34	3.47	0.00 **
Education	High school and Undergrad	Authoritative	0.01	0.06	0.02	0.14	0.89	0.10	0.06	0.20	1.75	0.08	−0.09	0.07	−0.14	−1.35	0.18
		Authoritarian	0.11	0.06	0.20	1.84	0.07	0.09	0.06	0.18	1.56	0.12	0.02	0.06	0.04	0.36	0.72
		Permissive	0.01	0.08	0.01	0.09	0.93	0.04	0.08	0.05	0.53	0.60	0.4	0.08	0.42	4.75	0.00 **
	Postgrad and PhD	Authoritative	0.02	0.06	0.04	0.30	0.76	0.03	0.07	0.05	0.42	0.67	−0.10	0.08	−0.16	−1.37	0.17
		Authoritarian	−0.01	0.06	−0.02	−0.12	0.90	0.11	0.07	0.19	1.52	0.13	−0.02	0.07	−0.03	−0.23	0.82
		Permissive	0.00	0.08	0.00	−0.01	0.99	0.06	0.09	0.07	0.72	0.48	0.23	0.10	0.22	2.40	0.01 **
Nationality	United States	Authoritative	0.09	0.09	0.17	0.96	0.34	0.00	0.08	0.01	0.04	0.97	−0.04	0.09	−0.07	−0.48	0.64
		Authoritarian	0.06	0.10	0.11	0.60	0.55	0.06	0.09	0.12	0.64	0.53	0.06	0.11	0.08	0.53	0.60
		Permissive	−0.02	0.11	−0.03	−0.20	0.84	−0.05	0.10	−0.08	−0.56	0.58	0.34	0.11	0.38	3.04	0.00 ^#^
	Asian	Authoritative	−0.21	0.12	−0.38	−1.74	0.09	0.17	0.13	0.29	1.35	0.19	−0.07	0.12	−0.13	−0.57	0.57
		Authoritarian	0.07	0.10	0.16	0.72	0.48	0.08	0.11	0.16	0.73	0.47	−0.08	0.10	−0.18	−0.8	0.43
		Permissive	−0.08	0.15	−0.10	−0.53	0.60	−0.04	0.16	−0.04	−0.23	0.82	0.04	0.16	0.05	0.28	0.79
	United Kingdom	Authoritative	0.10	0.07	0.21	1.34	0.18	0.08	0.08	0.15	1.00	0.32	−0.13	0.09	−0.20	−1.48	0.15
		Authoritarian	0.08	0.07	0.18	1.11	0.27	0.04	0.08	0.07	0.47	0.64	0.08	0.09	0.13	0.90	0.37
		Permissive	−0.09	0.11	−0.11	−0.79	0.44	0.17	0.12	0.18	1.38	0.18	0.44	0.14	0.40	3.10	0.00 ^#^
Kids	Yes	Authoritative	0.04	0.07	0.07	0.49	0.63	0.15	0.08	0.30	1.99	0.05 *	0.01	0.08	0.02	0.18	0.86
		Authoritarian	0.08	0.07	0.15	1.04	0.30	0.15	0.08	0.26	1.83	0.07	0.11	0.08	0.17	1.46	0.15
		Permissive	−0.04	0.10	−0.06	−0.45	0.66	0.05	0.10	0.06	0.51	0.61	0.34	0.10	0.36	3.31	0.00 *
	no	Authoritative	0.02	0.05	0.04	0.36	0.72	0.02	0.06	0.03	0.28	0.78	−0.13	0.07	−0.2	−1.95	0.05
		Authoritarian	0.07	0.05	0.14	1.22	0.23	0.07	0.06	0.13	1.21	0.23	−0.04	0.07	−0.06	−0.57	0.57
		Permissive	0.00	0.07	0.00	−0.01	1.00	0.05	0.08	0.06	0.65	0.52	0.32	0.08	0.32	3.77	0.00 **
Dog Caring Experience	Main Caregiver	Authoritative	−0.05	0.07	−0.09	−0.67	0.50	0.05	0.07	0.10	0.74	0.46	−0.09	0.08	−0.14	−1.11	0.27
		Authoritarian	0.03	0.08	0.06	0.45	0.66	0.09	0.08	0.15	1.07	0.29	0.07	0.08	0.10	0.84	0.40
		Permissive	0.00	0.09	0.00	0.04	0.97	0.04	0.09	0.05	0.48	0.63	0.33	0.09	0.35	3.47	0.00 **
	Family Dog	Authoritative	0.03	0.09	0.06	0.38	0.70	0.17	0.09	0.30	1.98	0.05	0	0.10	0	0.03	0.98
		Authoritarian	0.12	0.09	0.23	1.35	0.18	0.23	0.09	0.40	2.56	0.01 *	0.13	0.10	0.20	1.29	0.20
		Permissive	−0.04	0.11	−0.06	−0.39	0.70	0.10	0.11	0.11	0.85	0.40	0.24	0.14	0.24	1.79	0.08
	First Time Caregiver	Authoritative	0.06	0.08	0.12	0.75	0.46	−0.03	0.07	−0.05	−0.34	0.74	−0.16	0.08	−0.26	−1.86	0.07
		Authoritarian	0.05	0.07	0.11	0.65	0.52	−0.09	0.07	−0.18	−1.17	0.25	−0.09	0.08	−0.17	−1.18	0.24
		Permissive	−0.13	0.14	−0.16	−0.93	0.36	0.13	0.13	0.15	1.07	0.29	0.47	0.14	0.47	3.40	0.00 *

Note B = unstandardized regression coefficients; SE B = unstandardized regression coefficients standard error; ß = standardized regression coefficients. Bonferroni-adjusted p-interaction ^#^ *p* < 0.0167 for nationality; * *p* < 0.05; ** *p* < 0.01 for having kids in family.

**Table 4 animals-14-01038-t004:** Demographic information of participants in qualitative study.

	Gender	Age	Nationality	Education	Child/ Childless	Dog Caring Experience	Dog Breed	Acquisition	Caregiving Duration
Participant 01	Female	25–34	Asian	High School	No	First Time Caregiver	Mixed Breed	Rescue	6 years
Participant 02	Female	45–54	South African	Postgraduate	No	Experienced Caregiver (Family Dog)	1. Labrador cross-border collie 2. Belgian Malinois	Rescue Breeder	10 years 5 years
Participant 03	Female	35–44	Asian	Postgraduate	No	First Time Caregiver	Mixed Breed	Resue	4.5 years
Participant 05	Female	35–44	Canadian	Undergraduate	No	Experienced Caregiver (Main Caregiver)	Belgian Mailinos	Rescue	11.5 years
Participant 06	Female	25–34	Asian	Undergraduate	No	First Time Caregiver	Corgi	Adopted From Friend	7 years
Participant 07	Female	25–34	Asian	Postgraduate	No	Experienced Caregiver (Family Dog)	Sheep Dog	Adopted From Friend	11 or 12 years
Participant 08	Female	55+	British	Postgraduate	No	Experienced Caregiver (Main Caregiver)	Cockapoo	Breeder	12 years
Participant 09	Female	55+	British	High School	Yes	Experienced Caregiver (Main Caregiver)	Mixed Breed Mixed Breed	1. Rescue 2. Rescue	X
Participant 10	Female	25–34	American	Postgraduate	No	Experienced Caregiver (Main Caregiver)	Border collie	Breeder	1.5 years
Participant 11	Female	25–34	Asian	Postgraduate	No	Experienced Caregiver (Main Caregiver)	Mixed Breed	Rescue	3 years

Note: Participant 4 withdrew from the study; as a result, their data was excluded from the analysis. X = Participant 9 did not provide information about the caregiving duration during the interview.

**Table 5 animals-14-01038-t005:** The impact of received parenting behaviors on dog-directed parenting behaviors: themes, subthemes, and definitions.

Theme and Subtheme	Definition
Exploring the Complex Interplay of Human–Dog Relationship: Child, More Than Human, or just “Dog.”	The multifaceted roles dogs play within families and the influence of dogs’ role in the human–dog relationship impacts caregiving behavior.
1A Blurring the Human–Dog and Parent–Child Relationship Boundary	Perceived similarities between caring for dogs and raising children, emphasizing the emotional connections and kinship-like practices observed in caregiver–dog interactions.
1B More-than-Human Practices: Insights from The Increased Knowledge of Dog and Experienced Childhood Loneliness	Viewing dogs as a species with their own needs, transcending conventional pet caregiving practices, prioritize their bond with dogs and integrate them into their daily routines, adjusting their lifestyles to meet their dogs’ needs.
1C Situations Where a Clear Boundary Between Human–Dog and Parent–Child Relationships Exists: When Human or Dog cannot Meet One’s Own Expectations	Recognizing distinctions between caring for dogs and raising children, prompts adjustments in caregiving approaches, yet kinship practices persist.
Continuity, Discontinuity, and Compensation: Self-Reported Influence of Childhood Parenting Behaviors on Dog-Directed Parenting Behaviors	Reflections on received parenting behaviors in childhood influence caregiving behaviors on dogs
2A Continuity of Parenting Behaviors and Attitudes with High Responsiveness	Experiencing high levels of responsive parenting during childhood, along with positive parental attitudes towards non-human animals, leads individuals to replicate these behaviors towards their dogs.
2B Discontinuity of High-Control Parenting Behaviors and Attitudes	Experiencing high levels of demanding parenting behaviors that elicit negative emotions during childhood leads individuals to reject adopting similar parenting behaviors towards dogs.
2C Compensating for Low-Responsive Parenting Behaviors and Attitudes	Experiencing low levels of responsive parenting behaviors leads to compensatory parenting behaviors, displaying greater affection towards dogs even when perceived as not beneficial.

## Data Availability

Data are contained within the article and Appendix A, Appendix B, Appendix C, Appendix D and Appendix E.

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
