# Peer review of "Intergenerational Transmission of Human Parenting Styles to Human–Dog Relationships"

_animals, 2024, doi:10.3390/ani14071038_

Round 1

Reviewer 1 Report

Comments and Suggestions for Authors

The paper covers a topic of interest to many people working on human-dog relationship. However, some improvement could/should be made:

please avoid the terms "owner" and "animals". No sentient being should be the property of someone, and from a zoological point of viwew humans are also animals, thus "non-human animals" would be better

In the title, the DOG should be mentioned already

Some references should be made to

- the one health/one welfare approach

- the publications using LAPS for determining humnan-pet relationships

- in the discussion we also should have both the dogs' and society's point of view, not only the caregiver. Especially as eg Blachwell et al discusses the risk of unwanted behaviours in public as related to training methods.

- also the fact that "parenting style" or "interaction style" was practically unkmown in human-pet relationships before about 2014, the search trem "training method/style" should also be applied

- Schöberl, Kotrschal and colleagues in Vienna and Prado-Previde found differences betqween humnan-dog relationships between men & women. This should be discussed eg line 996 ff

- the questionnaire should be put into the supplement

in tab 1, the age, breed etc of the relevant dog might be of interest

- the method described in lines 390 ff is not familiar to non-humanities readers and should be explained, maybe by an example in the supplem

- fig 3 and similar statemebnts: It would be helpful to know which questions relate to what type of orientation

- some typos and formatting problems need to be addresse: eg l 46 ff, the sentence seems incomplete, l 160: martial or marital?l 400 - 403 duplicates.., the ref to Baumrind l 1211 is incomplete...., van Herwijnen is cited twice under "H" and "v"

ents

- E

Reviewer 2 Report

Comments and Suggestions for Authors

This topic of this paper is novel and would be a beneficial addition to the body of existing dog welfare published research.  It could also have numerous implications for the field of dog welfare.  It is very evident that the authors undertook a very scientifically rigorous study, and the usage of statistical analysis adds to the rigor and robustness of the paper.  Having said that, because the paper is quite long and rather dense, it might make more sense to divide the manuscript into two papers, with one focusing on the quantitative aspects of the study and the other focusing on the qualitative aspects.  All the comments and suggested edits are below, but one particular area that needs attention is in the Methods section regarding the adaptation of a questionnaire.  If the questionnaire was adapted as part of this study, this process needs to be thoroughly described, which would of course make the manuscript even longer.  And that would be an additional reason to divide the paper into two.

Abstract

Line 13:  the term “parenting” (or variations of the word) is commonly used in a casual context to refer to caring for a pet, but it does have connotations that might not be as appropriate for a scientific, although the term is used in other studies, and more formal context such as this, so using a different term such as “caregiving” would be ideal.

Personal reflective statement

This seems a bit of an odd section to include in the manuscript, and scientific papers in this field do not typically have such a section.

Introduction

Although there are high rates of pet keeping across many countries, cultures, etc., there are nonetheless differences in how pets are kept and the roles they play in particular societies and communities, such as community owned dogs, which could still be classified as owned and as pets, but nonetheless, there may well be differences in the dog-owner relationship compared to those owned by a single family.  It would be worthwhile to acknowledge that there are such differences in pet keeping.

Line 147:  typo – should be “impact”, not “impacts”

Line 210:  there seems to be typo – should probably be “parenting styles remain relatively unknown”

Line 213:  is study 1 referring to this study?  

Line 214:  these appear to be the study’s aims, not its hypotheses. And as such, it might be worthwhile to reword the following sentences to reflect what the aims of the study were, i.e. what was the study seeking to investigate, such as whether there was intergenerational transmission between parents’ receiving parenting behaviors…

Materials and Methods

Line 243:  it would be more appropriate to title this subsection “Participant Recruitment”

Lines 245-246:  the date range that participant recruitment took place should be noted.

Line 248:  “eligibility” does not need to be capitalized.

Lines 248-249:  it would seem that a key part of the eligibility criteria is that participants have a dog, so that should be mentioned here, as well as if there were any additional criteria pertaining to the dog, such as having to have owned the dog for a minimum amount of time.  Also, was the study open to participants located anywhere in the world, or only in specific countries?

Lines 253-267:  the sample size and all demographic info in this paragraph should be noted in the Results section, not here.

Lines 268-269:  this should only state the procedure undertaken, not the number of participants who were willing to be interviewed – that goes in Results.  So, this sentence should say something like “This study randomly selected 10 participants from the portion of the sample who responded that they were willing to be interviewed.”  Also, the reason for choosing 10 participants in particular should be given briefly.

Table 1 needs to be moved to the Results section.

Table 1 has typos in the Dog Caring Experience column – should be “first time caregiver”

Lines 301-305:  if this questionnaire was adapted as part of this study, then there needs to be a much more detailed explanation of how the questionnaire was adapted, such as with expert panel input.  The questionnaire adaptation process should constitute a subsection of the Methods section in and of itself.  If the questionnaire was developed elsewhere as part of a different study, then that needs to be noted with references here.  Merely stating that the questionnaire was adapted is grossly insufficient.

Line 343:  see previous comment about using the term “hypothesis” here.

Lines 401-403:  this looks to be part of the manuscript template so it should be deleted.

Lines 406-410:  it might make more sense to list the mean values in decreasing order with the highest mean first.

Table 2:  this table is large and contains a lot of information, so it would be ideal if it could be horizontally oriented.

Line 460:  should be “were examined”.

Results

Qualitative data analysis:  it might be helpful to display the themes and subthemes in some type of visual map in order to provide an overview of how they relate to each other (such as in Griffin et al., 2020).

Lines 553-559:  direct quotes should either be in quotation marks and/or in italics.  Also, it might be worthwhile to edit the direct quotes for clarity and brevity by using square brackets, ellipses, etc. 

Line 598:  typo – should be “caregivers”

Lines 648-650:  this beginning of this sentence doesn’t make much sense and either contains typos or just needs to be reworded.

Line 649:  the type of anthropomorphism should be clarified, i.e. are they avoiding anthropomorphism that can be detrimental to animal welfare due to an inability to differentiate their own needs and emotions as themselves as human with their dog’s, or critical anthropomorphism that can be a useful tool to improve dog welfare.  The former would make sense in this context, but nonetheless, it would be worthwhile to make this differentiation since they can have very different effects on animal welfare, and thus, not all anthropomorphism is “bad”.

Line 690:  typo – should probably be “one’s own expectations”/

Line 748:  typo – should be “participants”.

Lines 766-769:  this sentence doesn’t make a lot of sense – check or typos and consider rewording.

Lines 800-801:  this sentence doesn’t make sense.

Line 878:  typo – should be “Gilmore’s”.

Lines 904-906:  the Hemsworth (2015) reference isn’t especially relevant here, as even though horses are also considered pets, there are fundamental differences between horses as pets vs dogs as pets.

Line 928:  typo – should be “who observed”.

Lines 997-1003:  the first and second points are one in the same and should be combined, i.e. the participants were mostly female in this study, and thus greater effort should be made in future studies to include more male participants. The second point should instead be the final sentence of this paragraph – that future research should strive to have more culturally diverse samples.

Lines 1007-1009:  there is also an abundance of research that has concluded that pets do not positively impact human mental health, with some researchers in the field arguing that there is in fact more research that provides evidence for this than for how animals can improve humans’ mental health. 

There are line spacing and other formatting issues throughout the manuscript.

Were any dog demographics collected in the study?  If so, it would be interesting to investigate whether there’s a relationship between factors such as dog size and parenting style in particular, as studies have found that dog size is related to other areas of the dog-owner relationship, such as likelihood for relinquishment and dog training modality used.

Comments on the Quality of English Language

There are several typos throughout the paper, as well as a few non-sensical sentences that would benefit from being reworded.  Also, the direct quotes should be edited for clarity and brevity using ellipses, square brackets, etc.  In their current state, comprehension is challenging.  There are no major issues in this regard in the manuscript, and the minor issues that do exist are likely just oversights in the editing process and not a reflection of the overall English proficiency with which the manuscript was written.

Reviewer 3 Report

Comments and Suggestions for Authors

Dear authors:

This is a well written article, that includes important information. There has been a notable surge in the number of pet dogs and cats worldwide, and this is an aspect that has been taken great interest in recent years. Pets have taken on an increasingly important place in families, generating much closer bonds, resulting in a situation that emphasizes that dogs not only become incorporated into families but also actively participate in and influence everyday practices. This can establish a parallel between the parent-child relationship and the human-dog relationship. That is why this has been blurring the boundaries between human and animal relationships and it is a topic that you analyzed deeply on this document. However, I have added some comments and suggestions that I think could help improve the document.

Line 8: I understand that this journal template includes a section called Simply Summary that is missing in this document. Please add it before the Abstract section.

Line 13: Please add the missing space after “dog.”.

Line 255: Please add the missing point after “analysis”.

Line 253-267: I understand that you have carried out the analysis of the surveys based on the people who agreed to answer it. However, I consider that it could be interesting to try to make in the future other document, or even include this information in this article if you have it, with a sample with the same number of women as men, and even compare the vision of said genders. In the same way, it would also be interesting to include people from Latin America, as well as various economic levels in the study since this could generate important variations in the level of human-dog relationship.

Line 371: Please erase the doble space after “style,”.

Line 991: Please erase the doble space after “(Fig 5)”.

References: Please unify references list.

Round 2

Reviewer 1 Report

Comments and Suggestions for Authors

there are some minor typos ( eg line 199, 223,  348,  2342

Thanks for attending to all suggestions!!

Reviewer 2 Report

Comments and Suggestions for Authors

The manuscript looks to be in much better order after the comprehensive revisions.  There are still several typos throughout.  I have listed those that I spotted below, but it would be worthwhile to have another read through the entire manuscript to ensure there are no others.  My only other feedback on this manuscript is that it might be useful to briefly discuss (in the Discussion) what the practical applications are for the findings of this study.  It does add to the existing body of literature in this area, which is noted, but expanding beyond that to include how the results could be used in the real world, such as to support or educate dog owners, would be ideal.

Line 71:  typo – should be either “caregiving for a dog” or “caregiving for dogs”

Lines 79 & 197:  typo – “behaviours” is misspelled in both places

Line 348:  typo – “their” is misspelled

Line 1116:  typo – “experience” is misspelled

Line 1281:  typo – “non-human” is missing a hyphen

Line 1738:  typo – should be “humans’” and “dogs’” (plural instead of singular possessive) 

Line 2342:  typo – missing space between “animals” and “and”

There are inconsistencies in British and American spellings of words throughout – check to make sure they’re consistent.

Comments on the Quality of English Language

See above - the quality is good overall with a last few typos and grammatical errors to be fixed.
